# Effect of Temperature on Life History and Parasitization Behavior of *Trichogramma achaeae* Nagaraja and Nagarkatti (Hym.: Trichogrammatidae)

**DOI:** 10.3390/insects11080482

**Published:** 2020-07-29

**Authors:** Modesto del Pino, Juan Ramón Gallego, Estrella Hernández Suárez, Tomás Cabello

**Affiliations:** 1Andalusian Institute for Research and Training in Agriculture, Fishery, Food and Organic Production (IFAPA), Málaga Centre, Cortijo de la Cruz s/n, 29140 Churriana, Málaga, Spain; modesto.pino@juntadeandalucia.es; 2Research Centre for Mediterranean Intensive Agrosystems and Agrifood Biotechnology (CIAMBITAL), Agrifood Campus of International Excellence (CEIA3), University of Almeria, Ctra. de Sacramento, s/n, 04120 La Cañada, Almeria, Spain; jgg436@ual.es; 3Canarian Institute for Agricultural Research (ICIA), P.B. 60, ES 38200 La Laguna, Santa Cruz de Tenerife, Canary Islands, Spain; ehernand@icia.es

**Keywords:** *Trichogramma achaeae*, *Tuta absoluta*, *Chrysodeixis chalcites*, Trichogrammatidae, egg parasitoid, biological control, two-sex life table, functional response

## Abstract

*Trichogramma achaeae* Nagaraja and Nagarkatti (Hymenoptera: Trichogrammatidae) is currently used as biological control agent for several lepidopteran pests. Knowledge of thermal requirements is essential to optimize its rearing procedures and inundative releases. The biological characteristics and two-sex life table parameters of *T. achaeae* were determined at five constant temperatures (15, 20, 25, 30, and 35 °C) using *Ephestia kuehniella* Zeller (Lepidoptera: Pyralidae) eggs. *T. achaeae* was able to develop and survive from 15 °C to 30 °C, but not at 35 °C. Temperature significantly affected the preadult development time and adult longevity, decreasing when temperature increased from 15 °C to 30 °C. Temperature significantly altered the sex ratio, being female biased between 15 °C and 25 °C. Age-stage, two-sex life table analysis indicated that net reproductive rate (*R*_0_) was highest at 20 °C. Both the intrinsic rate of increase (*r*) and finite rate of increase (*λ*) increased with increasing temperature, while the mean generation time (*T*) decreased significantly. In addition, functional response of *T. achaeae* was studied, being significantly affected by temperature and host egg density, displaying a Holling type-I at 15 °C and a Holling type-II at 25 °C. The relevance of these results is discussed for the use of *T. achaeae* as biological control agent.

## 1. Introduction

Global trade and climate change have led to an exponential increase in the introduction and establishment of numerous invasive insect pests to new geographic areas, producing substantial ecological and economic impacts [1]. Several non-native Lepidoptera species have been established in Europe in recent years, and some native species have expanded their range within the continent [2]. Among the invasive species, the range of the South American tomato leafminer, *Tuta absoluta* Meyrick (Lepidoptera: Gelechiidae) has expanded rapidly since its accidental introduction in eastern Spain in 2006 [3]. In a few years, this species has rapidly spread throughout Europe, Africa, the Middle East and parts of Asia, becoming a devastating pest to tomato production worldwide [3,4,5]. Besides, due to climate change, many moth pest species are changing their migration routes from North Africa or southern Europe, invading new geographical areas where they can optimally reproduce and survive [6]. In this sense, the golden twin-spot moth or tomato looper, *Chrysodeixis chalcites* (Esper) (Lepidoptera: Noctuidae) is an extremely polyphagous species that has increased its movement and spread within central and northern Europe, causing severe damage to horticultural and ornamental crops both in open fields and in greenhouses [7], which indicates its potential expansion to other areas globally [8].

The appearance of lepidopteran pests in newly invaded areas has been predominantly managed by extensive application of broad-spectrum and persistent insecticides [5,9], which have often proved rapidly ineffective due to the development of resistance [10] and a multitude of undesired side effects on non-target organisms [11,12]. Thus, development of integrated pest management (IPM) programs is needed to attempt minimize the damage caused by these pests below the economic injury levels, reducing consequently the use of insecticides [13].

Among these strategies, inundative biological control with egg parasitoids of the genus *Trichogramma* (Hymenoptera: Trichogrammatidae) has shown efficient results in controlling several lepidopteran pests in many agricultural crops and forests worldwide [3,14,15]. These parasitoids can be inexpensively produced in large numbers and easily released in fields or greenhouses [16], reducing both egg hatching and subsequent crop damage caused by larval feeding [14], thus restricting the application of pesticides and contributing to the conservation of environment and human health [17]. *Trichogramma achaeae* Nagaraja and Nagarkatti have been reported as a potential biological agent for controlling *T. absoluta* and *C. chalcites* eggs in the Palearctic region [18]. It is a generalist and arrhenotokous species originating from India, which parasitizes the eggs of 26 lepidopteran species belonging to 10 different families [19], but it currently has extensive worldwide distribution due to its possible accidental or voluntary dispersion [18]. This parasitoid has been purposefully introduced for the biological control of lepidopteran pests to various European, African, South and Central American countries [3,18] and has been recently recorded on Hawaii [20] and the Canary Islands [18,21]. In addition, *T. achaeae* is currently commercialized and used in some European and North African countries against different Lepidoptera species in more than 15 horticultural and ornamental crops [3,22]. Indeed, *T. achaeae* has been shown to be a highly efficient biological control agent of *T. absoluta* in some Mediterranean Basin countries when is used in combination with inoculative releases of the omnivorous predator *Nesidiocoris tenuis* (Reuter) (Hemiptera: Miridae) in commercial tomato greenhouses [3,23,24,25]. Moreover, experimental inundative releases of *T. achaeae* have showed promising control results of *C. chalcites* in banana crops of the Canary Islands when they are combined with pest monitoring and additional application of biological insecticides as the entomopathogenic bacteria *Bacillus thuringiensis* var. *kurstaki* [13]. Additionally, *T. achaeae* has also been successful released against *Helicoverpa armigera* Hübner (Lepidoptera: Noctuidae) and other bollworms species in cotton crops in India [26].

The successful selection and introduction of *Trichogramma* wasps in biological control programs is determined by some important factors such as the potential development of the parasitoid species or strain in the target host, beside some abiotic and physical factors [27,28,29]. Among these, temperature is the most important environmental factor that affects the distribution and abundance of *Trichogramma* [30], because it strongly determines biological parameters such as the development rate, survival, longevity, parasitism rate, viability, sex ratio, and emergence rate of the parasitoid wasps [31,32]. Understanding the thermal requirements of a selected *Trichogramma* species/strain is also essential for planning its mass rearing procedures in laboratory [30,33] and to determine its potential as a biological control agent for a given pest and region [31]. However, the temperature effects on the biological parameters and thermal requirements of *Trichogramma* depends on species/strains studied [34,35] and the host used [31,36].

Another essential factor determining the selection and efficiency of a *Trichogramma* species as biological control agent is the number of parasitized eggs in response to the parasitoid and host densities, i.e., its functional response [37], defined by Solomon [38] as the relationship between the number of preys consumed by a predator individual and the prey density. Functional response is a basic element to understand the host-parasitoid interactions and has a prominent influence on the stability of the system [39]. Following Holling [40], the general form of the functional response has been categorized into three types (Holling’s type I, II, and III), according to the shape of the response curve below the upper limit. A type I response is characterized by a linear increase of killed host/prey to a plateau, a type II response by a curvilinear rise to a plateau that then levels off under the influence of handling time or satiation, and a type III response by a sigmoidal increase in hosts attacked [41,42]. In this sense, functional response is crucial for implementation of rapid population control required in inundative biological control [42]. Most *Trichogramma* species usually present a type I or type II functional response [42,43,44,45,46], but some studies have reported changes to type III because of the temperature [47,48] or the host typology [49].

Despite being a high economic importance species, the biology of *T. achaeae* is poorly studied [50,51,52,53], and no studies have been carried out in the past to determinate the variation on its life table parameters and functional response in relation to temperature, except its response to low temperatures for storage purposes [53,54,55] and improving its efficiency to control of *T. absoluta* [56]. Therefore, the aim of this work was to determine the life-cycle parameters and thermal requirements of *T. achaeae*, besides establishing its functional response in relation to temperature in *Ephestia kuehniella* Zeller (Lepidoptera: Pyralidae) eggs as an alternative host in order to optimize the mass production systems and inundative releases in the field of this parasitoid species for the biological control of *C. chalcites* and *T. absoluta*, among other lepidopteran pests.

## 2. Materials and Methods

### 2.1. Insect Rearing

A laboratory colony of *T. achaeae* was established from parasitized eggs of *C. chalcites* originally collected on banana crops in the locality of El Remo, La Palma, Canary Islands (Lat. 28°33′21″ N, Long. 17°53′17″ W). Emerging wasps were successfully identified by examination of morphological characters and molecular characterization by sequencing of the ITS2 gene [18,21]. *T. achaeae* was reared for at least 15 generations before starting the trials, using fresh ultraviolet-sterilized eggs of *E. kuehniella* as factious host in a climatic chamber (25 ± 1 °C, 60–80% RH and 16:8 h L:D photoperiod), according to the methodology proposed by Cabello et al. [23]. Factious host eggs were obtained from a culture of *E. kuehniella* massively reared following the methodology defined by Cerutti et al. [57]. A single layer of eggs of the factitious host were attached to a piece of paper, and the egg cards were offered to the adults of the parasitoid in glass vials (9.5 cm large × 1.5 cm diameter), covered with cotton, and some droplets of honey–water solution (1:1) as a food source. Parasitized eggs were preserved into the glass vials until the emergence of the adult parasitoids. According to Cabello and Vargas [58], before any experiment was conducted, the parasitoids were reared for three generations at each temperature to eliminate any effects due to previous temperature treatment.

### 2.2. Egg to Adult Developmental Time

The egg to adult developmental time of *T. achaeae* was studied at 15, 20, 25, 30, and 35 ± 1 °C, with a 60–80% RH and a 16:8 h L:D photoperiod in climatic chambers. From each temperature, newly emerged pairs (females and males) of *T. achaeae* were randomly chosen from the laboratory colony and exposed to fresh ultraviolet-sterilized eggs of *E. kuehniella* in small glass vials (5 × 1 cm). After 24 h, all parasitoids were removed, and 100 parasitized eggs (<24 h old) were individually isolated in glass vials with help of a wet fine brush (no. 0) and transferred to temperature-controlled chambers. Each isolated egg represented a replicate. Parasitized eggs were daily examined until the parasitoids completed their development and the sex of emerging wasps was recorded. Because larval development of *T. achaeae* occurs within the host egg, the immature stages (egg, larva, and pupa) were regarded as preadult stage (egg–pupa).

### 2.3. Adult Longevity and Fecundity

The longevity and fecundity of *T. achaeae* adults were also evaluated at 15, 20, 25, 30, and 35 ± 1 °C, with a 60–80% RH and 16:8 h L:D photoperiod. For each temperature, 20 male/female pairs less than one day old (<24 h) obtained in the development studies were separately isolated in small glass vials covered with cotton and a water–honey (1:1) drop for feed. Cardboards (2 × 2.5 cm) with approximately 100 fresh irradiated *E. kuehniella* eggs were supplied daily until the natural death of the adults. Parasitized host eggs were removed daily and incubated at the above-mentioned conditions until the offspring emergence. Adult longevity, fecundity (number of eggs parasitized per female and day), as well as the emergence rate (number of individuals emerged), and the sex ratio of the progeny (% female) were recorded.

### 2.4. Functional Response

The functional response of *T. achaeae* was determined at 15, 25 and 35 ± 1 °C, with a 60–80% RH and 16:8 h L:D photoperiod following the methodology described by Cabello et al. [46]. In all cases, newly emerged (less than 24 h old) adult and mated females of *T. achaeae* without previous parasitation experience, and *E. kuehniella* eggs (less than 24 h old) were used. For each test (15, 25, and 35 °C), the experimental design was totally randomized with only one factor—host density at five levels (10, 30, 50, 70 and 90 *E. kuehniella* eggs), carrying out 10 repetitions for each level of treatment. One mated female of *T. achaeae* was used per replicate. Host eggs were attached using a wet fine brush to cardboards (3 × 1 cm) in rows and columns separated 2 mm. Adult females were individually isolated in small glass vials and offered an egg cardboard for 24 h. Subsequently, the parasitoids were eliminated, and the egg cardboards were evolved up to offspring emergence. The registered data were parasitized eggs, emerged host larvae, and collapsed eggs.

### 2.5. Statistical Analysis

Development time, fecundity, and adult longevity data under different temperature regimes were transformed to log10(*x* + 1), while emergence rate and sex ratio data were transformed to arcsine (√ (*x*⁄100)). All transformed data were analyzed by ANOVA test applying the GLM procedure and the average values were compared by Tukey’s test (*p* = 0.05) by means of the statistical software IBM^®^ SPSS^®^ Statistics Version 22 [59]. The thermal constant and lower threshold temperature for development was calculated according to the linear thermodynamic model described by Ikemoto and Takai [60],
*DT* = *k* + *tD*(1)
where *DT* is the product of development time in days (*D*) and temperature (*T*) in degrees Celsius (°C). The intercept, *k*, is the thermal constant in degree days (°d), and the slope, *t*, the lower development threshold in °C. The parameters *t* and *k* were determined by linear regression.

The raw data on *T. achaeae* life history reared on *E. kuehniella* eggs at above temperatures were analyzed in accordance with the age-stage, two-sex life table theory [61,62], by means of the software TWOSEX-MSChart^®^ [63], and the results were plotted with SigmaPlot^®^ Version 14.0 (Systat Software, Inc., San Jose, CA, USA). Life table parameters were calculated including age-stage specific survival rate (*s_xj_*) (where *x* is age in days and *j* is stage), age-stage specific fecundity (*f_xj_*), age-specific survival rates (*l_x_*), age-specific fecundity (*m_x_*), life expectancy (*e_xj_*), reproductive value (*v_xj_*) and the intrinsic rate of increase (*r*), finite rate of increase (*λ*), gross reproductive rate (*GRR*), net reproductive rate (*R*_0_) and mean generation time (*T*), according to earlier theories [61,64]. Bootstrap technique with 100,000 resamples [65] was utilized to calculate the means values, variances, and standard errors of the population parameters [66]. A paired bootstrap test was used for statistical analysis.

For the experiment concerning to the functional response of *T. achaeae* after adjustments data were submitted to two classes of statistical analysis. First, the density factor significance was determined by GLM analysis and the average values were compared by Tukey’s test (*p* = 0.05) by means of the statistical software IBM^®^ SPSS^®^ Statistics Version 22 [59]. Second, the three equations of functional response (types I, II, and III) were then adjusted for *T. achaeae*, according to the following expressions [46,67,68]:(2)$$\mathrm{Type}\;I:\;N_{a}=N_{t}\left[1-exp\left(-a^{\prime }\cdot T\cdot P_{t}\right)\right]$$
(3)$$\mathrm{Type}\;\mathrm{II}:\;N_{a}=N_{t}\cdot \left[1-exp\left(-\frac{a^{\prime }\cdot T\cdot P_{t}}{1+a^{\prime }\cdot T_{h}\cdot N_{t}}\right)\right]$$
(4)$$\mathrm{Type}\;\mathrm{III}:\;N_{a}=N_{t}\cdot \left[1-exp\left(-\frac{\alpha \cdot T\cdot N_{t}\cdot P_{t}}{1+\alpha \cdot T_{h}\cdot N_{t}+\alpha \cdot T_{h}\cdot N_{t}^{2}}\right)\right]$$
where *N_a_* is the number of parasitized hosts; *N_t_* is the density of the host or prey; *a’* is the instantaneous search rate (equivalent to Nicholson-Bailey’s “area of discovery”: *a* = *a’T*, days^−1^); *T* is the total available search time (days); *P_t_* is the number of parasitoids; *T_h_* is the host-manipulation time; and *α* is parasitoid mortality potential.

The cited adjustments were performed by means of the software Tablecurve 2D, version 5.0 [69]. The corrected Akaike information criterion (AICc) was applied to select the best adjustment model, as it offers better statistical precision for comparisons between models than the regression coefficient (*r*^2^) [70]. However, the latter was calculated to determine the goodness of each performed adjustment.

## 3. Results

### 3.1. Egg to Adult Developmental Time

*T. achaeae* was able to develop and emerge between 15 °C and 30 °C on *E. kuehniella* eggs (Table 1). Statistical analysis manifested a significant influence of temperature on the egg–adult developmental time (*F* = 12,262.13, df = 3, *p* < 0.001). The developmental time was higher at lower temperatures, which ranged from 31.08 d at 15 °C to 7.55 days at 30 °C. In cultures maintained at 35 °C only emerged some adults of *T. achaeae*, which showed poor development of the wings and longevity less than 24 h, and none of the *E. kuehniella* offered eggs were parasitized, indicating that 35 °C exceeded the upper threshold. The egg-adult developmental time was not significantly different between males and females at the tested temperatures.

Developmental rate of *T. achaeae* fit the linear model suggested by Ikemoto and Takai [60]. The lower threshold temperature (*t*) was estimated in 10.51 ± 0.51 °C, while the thermal constant (*k*) was 138.27 ± 9.27°d (*r*^2^ = 0.995, *F* = 419.02, df = 1, *p* < 0.001).

### 3.2. Adult Longevity and Fecundity

The effects of different temperatures on the adult longevity, fecundity, preoviposition and oviposition periods, percentage of adult emergence, and sex of offspring of *T. achaeae* for each temperature tested are given in the Table 1. Significant effects of temperature were recorded on longevity of females (*F* = 17.82, df = 3, *p* < 0.001) and males (*F* = 17.50, df = 3, *p* < 0.001). In both cases, the adult longevity increased with the decrease of temperature, the maximum of 15 °C with 12.75 ± 1.22 d and 9.12 ± 1.04 days for females and males, respectively. There were significant differences in the longevity of *T. achaeae* females as compared to their conspecific males at the same temperature (*t* = 5.456, df = 155, *p* < 0.001). Adult female longevity was longer than male longevity at every tested temperature (Table 1).

The number of parasitized eggs per female (fecundity) was influenced by temperature and revealed significant differences between the tested temperatures (*F* = 15.15, df = 3, *p* < 0.001). The mean number of parasitized eggs were highest at 20 °C (64.05 ± 4.57 eggs/female) and lowest at 15 °C (28.45 ± 2.83 eggs/female) with intermediate values in the remaining temperatures. The adult preoviposition period (APOP) and total preoviposition period (TPOP) of *T. achaeae* were significantly different for the tested temperatures (*F* = 18.98, df = 3, *p* < 0.001 for APOP and *F* = 2232.46, df = 3, *p* < 0.001 for TPOP). No APOP was observed for the female parasitoids emerged at different temperatures tested, except at 15 °C (2.15 ± 0.49 d). The females developed at 30 °C showed the shortest TPOP (7.66 ± 0.10 d), but the highest was detected at 15 °C (33.15 ± 0.49 d). Temperature also significantly influenced the mean oviposition period of *T. achaeae* (*F* = 9.48, df = 3, *p* < 0.001), with the maximum at 15 °C (10.95 ± 1.13 d) and minimum at 30 °C (5.70 ± 0.37 d).

The percentage of adult emergence (fertility) was also significantly affected by temperature (*F* = 19.94, df = 3, *p* < 0.001). The highest adult emergence rate occurred at 30 °C (95.43 ± 0.64%), while the lowest was still at 15 °C (77.52 ± 2.21%). A female-biased sex ratio was recorded for *T. achaeae* under the temperature regimes studied. There was a significant difference in the sex ratios (*F* = 26.81, df = 3, *p* < 0.001) at tested temperatures. The values ranged from 48.41 ± 0.91% to 68.10 ± 1.90% females at 30 °C and 15 °C, respectively.

### 3.3. Age-Stage, Two-Sex Life Table of T. achaeae

According to the age-stage, two-sex life table theory, population parameters of *T. achaeae* were projected using the data of the entire cohort. Based on our results, population parameters exhibited significant differences over the rearing temperature (Table 2).

The greatest value of the net reproductive rate (*R*_0_) was observed at 20 °C (32.00 ± 5.52 eggs per individual) and the lowest at 15 °C (14.08 ± 2.59 eggs per individual). The intrinsic rate of increase (*r*) of the parasitoid also increased as temperature increased, recording the lowest value at 15 °C (0.072 ± 0.005 d^−1^) and the highest at 30 °C (0.319 ± 0.017 d^−1^). The highest value for the finite rate of increase (*λ*) occurred at 30 °C (1.38 ± 0.02 d^−1^) because of the shorter duration of one generation. Finally, the mean generation time (*T*) decreased with increased temperature and ranged from 10.36 ± 0.10 d at 30 °C to 36.71 ± 0.39 d at 15 °C.

The age-stage-specific survival rate (*s_xj_*) indicates the probability that an individual (as a newly laid egg) of *T. achaeae* will survive to age *x* and develop to stage *j*, depending on temperature is shown in Figure 1. These curves also show the survivorship and stage differentiation as well as the variable developmental rates. The survival rate of *T. achaeae* was highest at 15 °C and lowest at 30 °C.

The age-specific survivorship (*l_x_*), age-stage specific fecundity (*f_x_*), and age-specific fecundity (*m_x_*) of *T. achaeae* reared on *E. kuehniella* eggs at different temperatures are illustrated in Figure 2. They indicate that *T. achaeae* can successfully survive and reproduce on *E. kuehniella* eggs between 15 °C and 30 °C. The curves of age-specific survival rate (*l_x_*) show the probability that a newly laid egg will survive to age *x*. The survival rate of *T. achaeae* individuals at various temperatures was different. While the longest survival rate (53 d) occurred at 15 °C, its shortest value (17 d) was observed at 30 °C. The age-stage-specific fecundity (*f_xj_*) is the daily mean number of offspring produced by individual *T. achaeae* of age *x* and stage *j* per day. According to our results, the highest value of age-stage specific fecundity (*f_xj_*) of *T. achaeae* at 15, 20, 25 and 30 °C was 4.05, 22.40, 15.10 and 15.95 eggs per female and day, respectively, happening at the age of 32, 15, 9 and 8 d, respectively. The age-specific fecundity (*m_x_*) is the mean number of eggs produced per individual at age *x*. The age-specific maternity (*l_x_m_x_*) values changed depending on *l_x_* and *m_x_*, and the highest peak values of *l_x_m_x_* occurred at the age of 32, 15, 9 and 8 d at 15, 20, 25 and 30 °C, respectively. In accordance with the curves, the greatest fecundity happened in the two first days after female emergence.

Figure 3 shows the age-stage-specific life expectancy (*e_xj_*) of *T. achaeae*, which estimates the time that an individual of age *x* and stage *j* is expected to live. According to our results, the life expectancy of newly *T. achaeae* females was 12.75, 9.55, 7.55 and 6.25 d at 15, 20, 25 and 30 °C, respectively.

The curves of reproductive value (*v_xj_*) of *T. achaeae* are given in Figure 4. This parameter predicts the contribution of an individual of age *x* and stage *j* to the future population. Therefore, the reproductive value for the egg-pupa stage symbolizes the contributions of egg and pupal stages to the future population. The *v_xj_* for the egg-pupa stage of *T. achaeae* differed at temperatures tested, and adult female *v_xj_* peak was significantly highest at 20 °C but lowest at 15 °C, and it occurred earliest at 30 °C but latest at 15 °C. The highest adult female *v_x_*_j_ peaks significantly raised drastically to 20.05 at 31 d at 15 °C to 45.38 at 15 d at 20 °C to 34.14 at nine d at 25 °C and to 35.23 at eight days at 30 °C.

### 3.4. Functional Response of T. achaeae

Table 3 illustrates the mean number of parasitized eggs in the functional response test of *T. achaeae* at 15 °C and 25 °C, using *E. kuehniella* as host under laboratory conditions. The analysis of variance revealed a greatly significant effect of the host density on parasitism for each examined temperature (*F* = 17.04, df = 4, *p* < 0.001 at 15 °C and *F* = 19.65, df = 4, *p* < 0.001 at 25 °C). The greatest temperature (25 °C) can be observed to lead to higher quantities of parasitized eggs (average 16.78 eggs) than the lowest temperature (15 °C, average 12.70 eggs).

A type I functional response corresponded to the values of the number parasitized eggs (according to density) at 15 °C and a type II functional response at 25 °C (Figure 5), whose adjustments showed lower values in the corrected Akaike indices (AICc) (Table 4). The instantaneous search rate (*a’*) was maximum at 25 °C (11.834 h^−1^) and minimum at 15 °C (0.8097 h^−1^). Host-manipulation time was 0.0414 h at 25 °C and very low at 15 °C. At 35 °C, females of *T. achaeae* were able to survive for 24 h; however, the percentage of parasitism found was very low, so that these values were not allowed to adjust the functional response. Also, the further development of the offspring of females did not occur to any of the densities studied. In this case, the effect was caused by the high mortality of *E. kuehniella* eggs at 35 °C, which collapsed and dried even when parasitized.

## 4. Discussion

According to Samara et al. [71], temperature is considered the most significant abiotic factor directly influencing development rate, cumulative fertility, adult longevity, sex ratio and emergence rate of *Trichogramma*. In the present study, we evaluated the influence of five constant temperatures (15, 20, 25, 30 and 35 °C) on the biological characteristics and two-sex life table parameters of *T. achaeae* when it is reared in *E. kuehniella* eggs under laboratory conditions. Results in the current study clearly showed that temperature significantly affected the preadult development time, decreasing when temperature was increased from 15 °C to 30 °C. According to Bueno et al. [72], this shortening of development time is caused by an increase in the metabolic activity of the immature stages at higher temperatures. The tendency of variation between temperatures was similar for both sexes, and differences in developmental time were not observed between *T. achaeae* females and males at all tested temperatures. Similar results have been published by other authors in biological studies of different *Trichogramma* species [30,35,58,72,73]. In our study, the egg to adult development of *T. achaeae* at 25 °C was completed in 8.83 d in *E. kuehniella* eggs, lower than the 10.4 d and 10.8 d recorded by Ghosh et al. [51] and Manohar et al. [52] in *Corcyra cephalonica* Stainton (Lepidoptera: Pyralidae) and *T. absoluta* eggs, respectively. These differences might be associated with the effect of the size and nutritional quality of the host egg on the preimaginal development and adult fitness of *Trichogramma* spp., as suggested by Kishani Farahani et al. [74]. Our studies indicated that *T. achaeae* developed successfully from egg to adult in the temperature range of 15 °C to 30 °C. On the other hand, the parasitized host eggs turned black at 35 °C, but many of them collapsed, dried, and no viable progeny emerged, showing a negative effect of high temperature, thus exceeding the upper threshold. The high temperature preventing the adult emergence observed in our study agrees with previous works [34,58]. However, Ghosh et al. [51] found that *T. achaeae* was successfully able to develop and emerge at 36 °C in *C. cephalonica* eggs.

Knowledge of thermal requirements allows us to expect the number of generations of *Trichogramma* spp. and the best time for the release of the parasitoids in the field [31]. In the present work, temperature thresholds and thermal requirements of *T. achaeae* were calculated according to linear model suggested by Ikemoto and Takai [60]. Our results showed that the developmental threshold temperature (*t*) and the effective accumulated temperature (*k*) values for egg to adult were 10.51 °C and 138.27 °d, respectively. These values were close to those previously reported for *T. achaeae* [51] and for other subtropical *Trichogramma* species [35], suggesting a tolerance of this species to low temperatures. In this line, several authors indicated that pupae of *T. achaeae* can be successfully placed at 10 °C for short term storage up to 30 days, resulting in an adult emergence percentage of 60% at this temperature [53,55]. The time taken for *T. achaeae* to complete its development at 15 °C (31.08 d) shows that the development rate in preadult stages within *E. kuehniella* eggs is prolonged at lower temperatures. In this sense, earlier studies suggest that immature stages inside the host are the dominant overwintering stage of *Trichogramma* [35], that corroborates the theory that *T. achaeae* does not enter diapause, its development being slower through the winter months [3,56]. On the other hand, the absence of diapause in *T. achaeae* can cause difficulty in storage and handling, which, together with its biparental reproduction, makes the use of this parasitoid more difficult and expensive as biological control agent [24,56].

The current study showed that temperature significantly affected adult longevity, fecundity, parasite emergence, and sex ratio of *T. achaeae*. As observed in other *Trichogramma* species [27,30,75], the mean adult longevity of *T. achaeae* females and males decreased as temperature increased, recording the highest longevity at 15 °C, for both females and males. According to Foerster et al. [75], increase in adult longevity may be due to a reduction of parasitoid activity and metabolic rate at low temperatures. The results also showed that *T. achaeae* can reproduce through the studied temperature range of 15–30 °C on *E. kuehniella* eggs. The highest number of parasitized eggs or fecundity was found at 20 °C (64.05 eggs per female) and then decreased as temperature increased. These results are according with those found for other species as *T. cordubensis* Vargas and Cabello [58] and *T. aurosum* Sugonjaev and Sorokina [71], where the parasitized host eggs number increased with increasing temperature to a maximum at 20–25 °C and declined at 30 °C. The deleterious effect of high temperature on the fecundity may be caused by the reduction in female longevity [58]. On the other hand, consulted references have shown an important variation in the number of eggs parasitized by *T. achaeae* according to the rearing host. We found that females of *T. achaeae* parasitized 55.75 eggs of *E. kuehniella* at 25 °C, in line with the 58.4 eggs of *C. cephalonica* reported by Ghosh et al. [51], but higher than the 35.37 eggs of *E. kuehniella*, and the 22.13 eggs of *T. absoluta* found by Melo [76] and Manohar et al. [52], respectively. Likewise, it is known that the adult longevity and reproductive skills of *Trichogramma* spp. may change in accordance with the temperature [73] and host nutritional quality during the development [74] and food in the adult stage [77]. The fertility was maximum at the first 24 h after emergence of *T. achaeae* females, and no preoviposition period was detected at all tested temperatures, except at 15 °C. After the first day, the oviposition peak consistently declined until female death, being more notable at those temperatures in which fertility was highest. This reduction in the number of parasitized eggs over the female longevity increased is typical in pro-ovigenic species as the parasitoids of the genus *Trichogramma* [27].

Emergence rate or viability of *T. achaeae* was relatively stable at all tested temperatures. The emergence rate values, except for those observed at 15 °C, were always above 80%, being the highest viability measured at 30 °C (95.43%). These values are greater than the previous results found for *T. achaeae* in *C. cephalonica* eggs [51], indicating that the viability can be influenced by both host and temperature [72]. The reduction in the emergence rate recorded at the lowest temperature (15 °C) can be due to the high mortality in the immature stages [71]. A single parasitoid emerged per egg of *E. kuehniella*, however it was common to find up 2–3 emerging adults from *C. chalcites* eggs collected in banana crops [18,21]. These differences could be caused by the bigger size of *C. chalcites* eggs compared to *E. kuehniella* eggs and its nutritional quality for the parasitoid development [73]. Despite that, our findings clearly shown that *E. kuehniella* eggs can be considered an appropriate alternative host to support the development of *T. achaeae*. Temperature is also a strongly important factor in affecting the sex ratio in *Trichogramma* spp. [31,73]. However, the influence of temperature on the sex ratio differs according to the species/strain of *Trichogramma* [72]. In our study, temperature significantly modified the sex ratio of *T. achaeae*, being female biased between 15 °C and 25 °C, whereas it was balanced at 30 °C. Percentage of females fluctuated from 48.41 to 68.10, which is coherent with Melo [76], who noticed that sex ratio was biased to female production at the same temperatures.

In accordance with our knowledge, no studies have been carried out before to determine the temperature influence on population parameters of *T. achaeae*. Two-sex life tables are a powerful instrument to describe the probable development of a species [61]. Results obtained in this work indicate that *T. achaeae* was able to quickly increase its populations in the temperature range of 15 °C to 30 °C. Several authors have adopted fertility life tables for estimating potential of *Trichogramma* species or strains submitted to different factors such as temperature and host adaptability [31,78]. The present study reports significant variations in the two-sex life table parameters of *T. achaeae* in response to the distinct temperature tested. The net reproductive rate (*R*_0_) varied according to the temperature variation, from 14.08 to 32.00 offspring per individual. The highest increase in net reproductive rate was observed at 20 °C. The net reproductive rate was adversely influenced by temperatures below or above 20 °C, and therefore, the population growth capacity was particularly reduced. According to Cabello and Vargas [58], the diminution of the net reproductive rate at elevated temperatures may be due to the production of both males and females at these temperatures. Results also showed that intrinsic rate of increase (*r*) and finite rate of increase (*λ*) were proportionally related to temperature, increasing considerably until the highest temperature was reached, while the mean generation time (*T*) was lowest at 30 °C. Therefore, our results indicated that *E. kuehniella* is an appropriate host for the mass rearing of *T. achaeae*, and they are in line with those reported by Varma and Maninder [50] and Manohar et al. [52] on *C. cephalonica* and *T. absoluta* eggs, respectively.

Therefore, the determination of the biological characteristics, thermal requirements and population parameters of *T. achaeae* on *E. kuehniella* eggs contributed essential information for the development of biological control programs, because the results demonstrated that this parasitoid species is able to survival and develop within the tested temperature range (from 15 °C to 30 °C). In this sense, Cabello et al. [23] found that *T. achaeae* is a species well adapted to extreme temperature conditions in commercial tomato greenhouses in southern Spain. Our results would explain the presence of *T. achaeae* throughout the year in banana crops in the Canary Islands, both in mesh-built greenhouses and in open fields [21], where a Mediterranean-type climate is present and *C. chalcites* eggs are available for the parasitoid survival during all banana production cycle [8,79,80]. Thus, the thermal tolerance of *T. achaeae* in this study to both average minimum and maximum temperatures occurring in the Canary Islands (16–27 °C), make this species a suitable candidate as biocontrol agent of *C. chalcites* in banana groves [13]. However, extreme temperatures had a detrimental impact on the biology and parasitization rates of *T. achaeae* [56,76].

No experiments have been performed before to determine the relationship between host densities and functional response for *T. achaeae* at different temperatures. The current study revealed that temperature had a substantial effect on the functional response of the egg parasitoid. Several studies have observed that the functional response of *Trichogramma* could change from one type to another as temperature [47,81] and host typology change [49]. Besides, similar changes have also been reported in other parasitoid species [68]. According our results, *T. achaeae* showed a type I functional response at 15 °C. In this sense, Jeschke et al. [82] indicate that a type I response is attributable to the following two essential requisites: 1) a very short handling time (*T_h_*) and 2) a state of maximum activity at saturation. However, in laboratory conditions at 15 °C, both requisites seem to be met for *T. achaeae*, who seems to have a very low *T_h_*, and only ceases to lay eggs when the female reaches its eggs in the ovarioles. These findings agree with those seen for other *Trichogramma* and *Trichogrammatoidea* species, in which most cases present type I functional responses [42,43,44,45,46]. In the other hand, at 25 °C the functional response of *T. achaeae* is type II, which is influenced by the time of handling. Our results are in line with a type II functional response exhibited by *T. achaeae* to *T. absoluta* eggs [83]. However, it should be noted that, in a species of the genus, *T. ostrinae*, the functional response is type II but changes to type III when the temperature increases from 20 °C to 27 °C [47]. Besides temperature, another factor that seems to condition the type of functional response in parasitoids of the genus *Trichogramma* is the host species; thus *T. pretiosum* Riley parasitizing *T. absoluta* eggs presents a type I functional response [44], but when it parasitizes *H. armigera* eggs, the response is type II [81]. Also, *T. chilonis* Ishii that presents type II response in eggs from its natural host *Chilo sacchariphagus* (Bojer) changes to type III when offered eggs from its unnatural host *Galleria mellonella* L. [49]. On the other hand, the functional responses change to type II when *T. achaeae* females must parasitize previously parasitized eggs, which is an extremely common phenomenon under natural conditions.

## 5. Conclusions

This study is the first to report the effect of temperature on two-sex life table parameters and functional response of *T. achaeae*, an important parasitoid of *T. absoluta* and *C. chalcites* in the Paleartic region. The age-stage, two-sex life-table analysis obtained from this research has provided fundamental information on the effects of temperature on the developmental time, survivorship, reproduction, and longevity of *T. achaeae*. Results showed that temperature ranging from 15 °C to 30 °C was favorable for its growth, development, and survival, as well as the adequacy of *E. kuehniella* eggs as alternative host in *T. achaeae* mass rearing systems. In addition, the results showed a clear relationship between temperature and the functional response of *T. achaeae*. Parasitoid female had a greater efficiency at 25 °C, showing a Holling type II functional response. The rate of parasitism increased when the host density increased. The present study can facilitate the optimization of laboratory conditions for the mass rearing of *T. achaeae* and present a starting point to enhance the biological control of several lepidopteran pest species.

## Figures and Tables

**Figure 1 insects-11-00482-f001:**
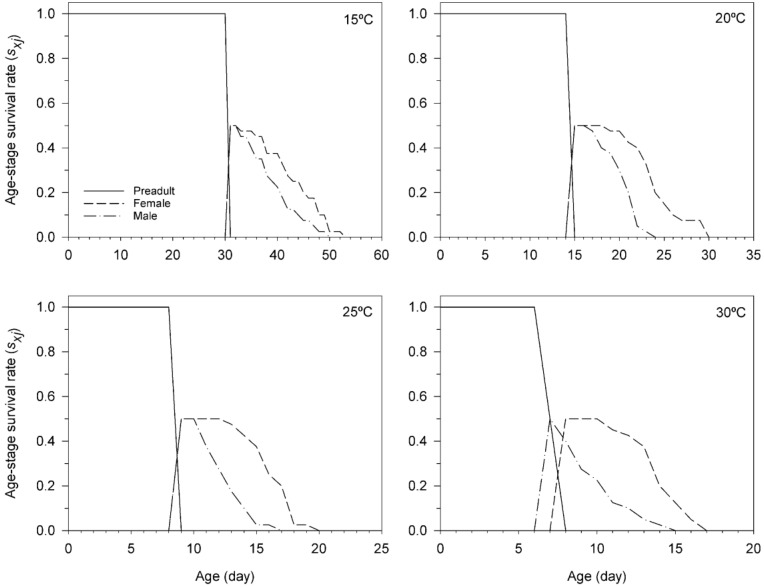
Age-stage survival rate (*s_xj_*) of *Trichogramma achaeae* reared at 15, 20, 25, and 30 °C on *Ephestia kuehniella* eggs, using the age-stage, two-sex life table.

**Figure 2 insects-11-00482-f002:**
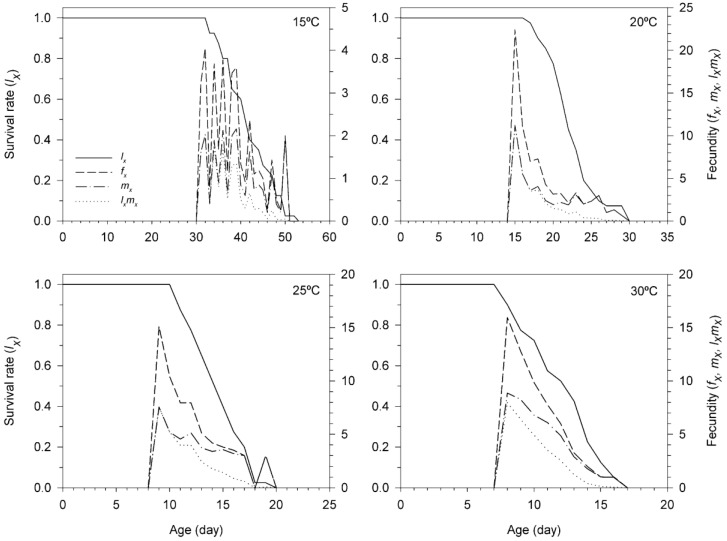
Age-specific survival rate (*l_x_*), age-stage specific fecundity (*f_xj_*), age-specific fecundity (*m_x_*) and age-specific maternity (*l_x_m_x_*) of *Trichogramma achaeae* reared at 15, 20, 25 and 30 °C on *Ephestia kuehniella* eggs, using the age-stage, two-sex life table.

**Figure 3 insects-11-00482-f003:**
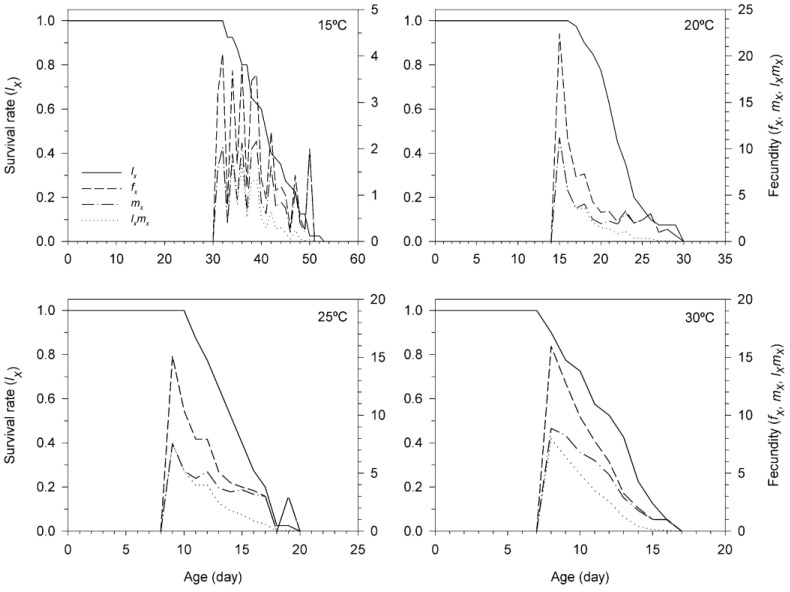
Age-stage-specific life expectancy (*e_xj_*) of *Trichogramma achaeae* reared at 15, 20, 25 and 30 °C on *Ephestia kuehniella* eggs, using the age-stage, two-sex life table.

**Figure 4 insects-11-00482-f004:**
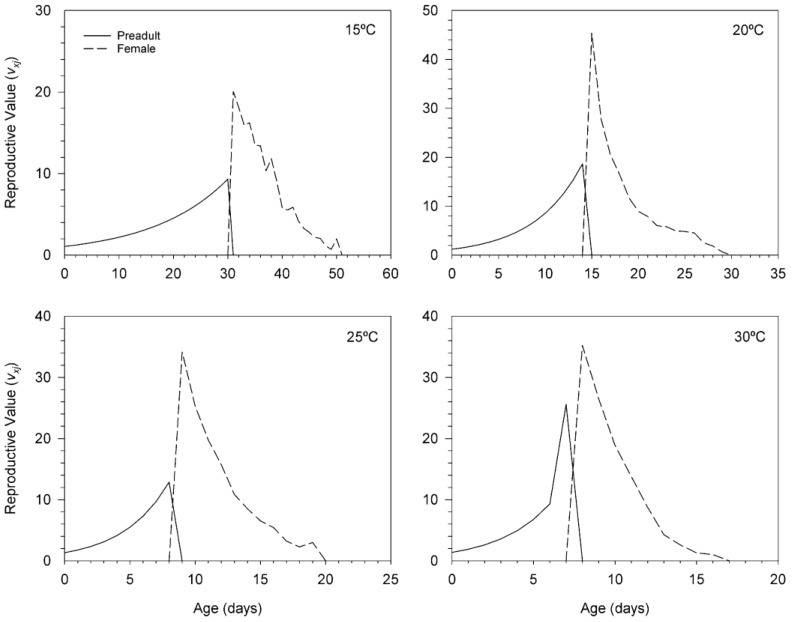
Age-stage reproductive value (*v_xj_*) of *Trichogramma achaeae* reared at 15, 20, 25 and 30 °C on *Ephestia kuehniella* eggs, using the age-stage, two-sex life table.

**Figure 5 insects-11-00482-f005:**
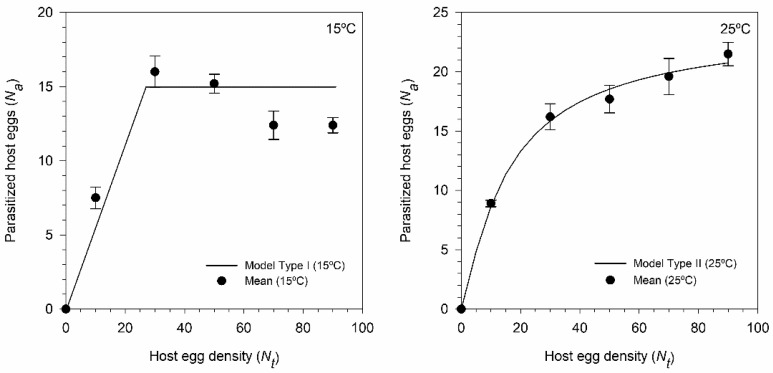
Functional response of *Trichogramma achaeae* female parasitizing *Ephestia kuehniella* eggs at two different temperature levels and under laboratory conditions (60–80% R.H. and 16:8 h L:D). (Vertical lines represent standard errors of models, *p* = 0.05).

**Table 1 insects-11-00482-t001:** Development time, adult longevity, adult preoviposition period (APOP), total preoviposition period (TPOP), oviposition period, fecundity, percentage of adult emergence, and sex ratio (mean ± SE) of *Trichogramma achaeae* reared on *Ephestia kuehniella* eggs at different temperatures under laboratory conditions (60–80% RH and 16:8 h L:D).

Parameters	Temperature (°C)			
	15	20	25	30
Total development time (days)	31.08 ± 0.14 a	14.52 ± 0.08 b	8.83 ± 0.05 c	7.55 ± 0.07 d
Female longevity (days)	12.75 ± 1.22 a	9.55 ± 0.67 b	7.10 ± 0.35 bc	5.55 ± 0.40 c
Male longevity (days)	9.12 ± 1.04 a	5.65 ± 0.42 b	3.95 ± 0.38 bc	3.40 ± 0.47 c
APOP (days)	2.15 ± 0.49 a	0.00 ± 0.00 b	0.00 ± 0.00 b	0.00 ± 0.00 b
TPOP (days)	33.15 ± 0.49 a	14.49 ± 0.11 b	8.82 ± 0.06 c	7.66 ± 0.10 d
Oviposition period (days)	10.95 ± 1.13 a	8.60 ± 0.69 ab	7.50 ± 0.39 bc	5.70 ± 0.37 c
Fecundity (eggs/female)	28.45 ± 2.83 b	64.05 ± 4.57 a	55.75 ± 3.60 a	54.30 ± 4.57 a
Emergence (%)	77.52 ± 2.21 c	86.31 ± 1.88 b	89.88 ± 1.65 b	95.43 ± 0.64 a
Sex offspring (% female)	68.10 ± 1.90 a	63.15 ± 1.66 ab	59.64 ± 1.69 b	48.41 ± 0.91 c
Sex offspring (% male)	31.90 ± 1.90 c	36.85 ± 1.66 bc	40.36 ± 1.69 b	51.59 ± 0.91 a
Sex ratio (F:M)	2.13:1	1.71:1	1.48:1	0.94:1

Means in each row followed by the same letter are not different using Tukey test (*p* < 0.05).

**Table 2 insects-11-00482-t002:** Population parameters (mean ± SE) of *Trichogramma achaeae* reared on *Ephestia kuehniella* eggs at different temperatures under laboratory conditions (60–80% RH and 16:8 h L:D).

Parameters	Temperature (°C)			
	15	20	25	30
*R*_0_ (offspring)	14.075 ± 2.591 c	32.000 ± 5.519 a	27.875 ± 4.756 b	27.150 ± 4.843 b
*r* (day^−1^)	0.072 ± 0.005 c	0.195 ± 0.010 b	0.284 ± 0.015 a	0.319 ± 0.017 a
*λ* (day^−1^)	1.075 ± 0.006 c	1.215 ± 0.012 b	1.328 ± 0.020 a	1.375 ± 0.024 a
*T* (day)	36.707 ± 0.397 a	17.761 ± 0.132 b	11.728 ± 0.156 c	10.357 ± 0.100 d
*GRR* (offspring)	23.070 ± 3.908 c	46.210 ± 6.713 a	42.280 ± 5.817 b	41.520 ± 5.861 b

*R*_0_: net reproduction rate. *r*: intrinsic rate of increase. *λ*: finite rate of increase. *T*: mean generation time. *GRR*: gross reproductive rate. Means in each row followed by the same letter are not different using paired bootstrap procedure (*p* < 0.05).

**Table 3 insects-11-00482-t003:** Number of *Ephestia kuehniella* eggs parasitized by *Trichogramma achaeae* females according to host density and temperature under laboratory conditions (60–80% RH and 16:8 h L:D).

Egg Density	Temperature (°C)		Average
	15	25	
10	7.50 ± 0.73 c	8.90 ± 0.28 c	8.20 ± 0.41
30	16.00 ± 1.06 a	16.20 ± 1.10 b	16.10 ± 0.75
50	15.20 ± 0.63 ab	17.70 ± 1.16 ab	16.45 ± 0.70
70	12.40 ± 0.96 b	19.60 ± 1.53 ab	16.00 ± 1.21
90	12.40 ± 0.96 b	21.50 ± 0.99 a	16.95 ± 1.18
Average	12.70 ± 0.55	16.78 ± 0.77	

Means followed by the same letter in the same column are not significantly different at *p* < 0.05 (Tukey test).

**Table 4 insects-11-00482-t004:** Parameters and statistical significance for functional response equations for number of *Ephestia kuehniella* eggs parasitized by *Trichogramma achaeae* females, at two temperatures (15 and 25 ± 2 °C) under laboratory conditions (60–80% RH and 16:8 h L:D).

Temperature (°C)	Type	Parameters ^1^				Statistical Parameters	
		*T_h_*	*a’*	*α*	AICc	df	*r* ^2^
	I	-	0.8097	-	4.0076	3	0.9342
15	II	0.0706	9.9374	-	11.259	5	0.7695
	III	0.0629	-	2.3843	11.440	5	0.7612
	I	-	0.8557	-	8.8033	3	0.8325
25	II	0.0414	11.834	-	−2.5839	5	0.9923
	III	0.0422	-	15.548	−2.3435	5	0.9901

^1^ Parameters: *a’*, instantaneous search rate (h^−1^); *α*, parasitoid mortality potential; *T_h_*, host-manipulation time (h).

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
