# Peer review of "Effect of Temperature on Life History and Parasitization Behavior of Trichogramma achaeae Nagaraja and Nagarkatti (Hym.: Trichogrammatidae)"

_insects, 2020, doi:10.3390/insects11080482_

Round 1
Reviewer 1 Report
The manuscript, “Effect of temperature on life history and parasitization behavior of Trichogramma achaeae Nagaraja & Nagarkatti (Hym.: Trichogrammatidae)” examined the life table of T. achaeae under five temperatures and effects on temperature on functional response. It is an important work toward developing biological control program for serious, economic pests such as Tuta absoluta. It is a well-written manuscript, understandable and data analyses seem to be appropriate. All the sections appear to be thorough with adequate details. The reviewer sees no flaws in the experimental design.
Minor comments:
Introduction
Line 46: change to “inexpensively”.
M&M
2.4 functional response: How many mated females were released to each replica?
Discussion
Line 395: Change to “consistently”.
Author Response
Response to Reviewer 1
Line number |
Changes |
L58 |
“inexpensive” changed to “inexpensively” |
L156 |
“One mated female of T. achaeae was used per replicate” added |
L395 |
“consistent” changed to “consistently” |
Reviewer 2 Report
The most important problem is that the novelty of the study is very limited. A number of similar investigations have been conducted with numerous insects including several Trichogramma species. On the other hand, the data obtained by the authors can be used in mass rearing of Trichogramma achaeae. Therefore the results of the present work are not of fundamental but definitely of applied value.
Second, it seems to me that there are some contradictions in the data:
- It is stated (line 243) that the lowest adult emergence rate occurred at 15°C (77.52 ± 2.21 %). On the other hand, in Fig. 1 it is clearly seen that age-stage survival of just emerged males and females at 15°C was close to 50% and was evidently higher than 40% which – if I do correctly understand the terms used by the authors – means that the rate of adult emergence was higher than 80%. Besides, in the Discussion (line 400) it is also stated that “emergence rate values, even with the difference observed at 15ºC, were always above 80%”.
- It is stated (line 279) that the highest value of age-stage specific fecundity at 15, 20, 25 and 30ºC was observed at the age of 8, 15, 9 and 8 days, respectively. On the other hand, it is stated (line 283) that the greatest fecundity happened in the two first days after female emergence that is certainly not at the age of 8 days at 15ºC. Moreover, the highest age-specific maternity was observed at the age of 32, 15, 9 and 8 days at 15, 20, 25 and 30ºC (the same is seen in Fig. 2). Thus, I guess that in the line 279 instead of “8, 15, 9 and 8 days” should be “32, 15, 9 and 8 days”.
The authors should be more careful with the numbers. Please, check all your data again.
Besides, I have a number of minor corrections, comments, and questions (see below).
Line 21: I guess, not “was available to develop” but “was able to develop”
Line 58: not “be inexpensive produced” but “be inexpensively produced”
Lines 69-70 not “more than 15 crops both horticultural and ornamental” but “more than15 horticultural and ornamental crops”
Table 1: if egg-adult developmental time was not significantly different between males and females (line 206), it is not reasonable to include in the table separate rows with these data. I think that the 3rd row (total development time) is enough.
Lines 213-219: Again, if the lower thermal threshold and the thermal constant of males and females are significantly different, please, indicate the level of significance. If not, there are no reasons to report these data separately.
Line 232: I guess, not “the mean parasitized eggs” but “the mean number of parasitized eggs”.
Line 239: not “influenced significantly in the mean oviposition period” but “influenced significantly on the mean oviposition period”
Table 3: only the difference between the numbers in the same column is indicated (line 313), although the difference between the data in the same row (the influence of temperature at a given host density) could be also interesting.
Line 315: not “corresponded in the values” but ““corresponded to the values”
Line 371: I would replace “does not present diapause” with “does not enter diapause” or “does not able to diapause”
Line 393: What is “The specific fertility”? This term has not been used above.
Lines 415-416: I think, not “to determinate the temperature influence” but “to determine the temperature influence”
line 444: not “extreme temperatures reported a detrimental impact” but “extreme temperatures have a detrimental impact” or “extreme temperatures were reported to have a detrimental impact”
Author Response
General comments:
- In our case, emergence rate values are calculated from the parasitized eggs obtained in the adult longevity and fecundity studies (M&M section 2.3.) and not from the parasitized eggs that were isolated in the egg to adult developmental time studies (M&M section 2.2.). In the Discussion, the line 394 “emergence rate values, even with the difference observed at 15ºC, were always above 80%” have been changed to “emergence rate values, except for those observed at 15ºC, were always above 80%”.
- The highest values of age-stage specific fecundity have been reviewed and changed in the text. Line 279: “According to our results, the highest value of age-stage specific fecundity (fxj) of achaeae at 15, 20, 25 and 30ºC was 4.05, 22.40, 15.10 and 15.95 eggs per female and day, respectively, happening at the age of 32, 15, 9 and 8 days, respectively.”
Line-specific comments:
Line number |
Changes |
L21 |
“was available to develop” changed to “was able to develop” |
L58 |
“be inexpensive produced” changed to “be inexpensively produced” |
L69-70 |
“more than 15 crops both horticultural and ornamental” changed to “more than15 horticultural and ornamental crops” |
Table 1 |
The egg-adult developmental time data for males and females have been removed, leaving only the values for all adults. Therefore, to ensure consistency between the text and the data shown in table 1, lines 199-205 and 334-336 have been rewritten. |
L213-219 |
The lower thermal threshold and the thermal constant values for males and females have been removed, leaving only the calculated values for all adults. Therefore, lines 356-358 have been rewritten. |
L232 |
“the mean parasitized eggs” changed to “the mean number of parasitized eggs”. |
L239 |
“influenced significantly in the mean oviposition period” changed to “influenced significantly on the mean oviposition period” |
L315 |
“corresponded in the values” changed to “corresponded to the values” |
L371 |
“does not present diapause” replaced with “does not enter diapause” |
L393 |
“The specific fertility” changed to “The fertility” |
L415-416 |
“to determinate the temperature influence” changed to “to determine the temperature influence” |
L444 |
“extreme temperatures reported a detrimental impact” changed to “extreme temperatures had a detrimental impact” |